# OpenReview forum: "Reward-free Alignment for Conflicting Objectives"
_ICML.cc/2026/Conference — ICML 2026 spotlight_

### Official Review · Reviewer_yLW1 · 2026-03-11

**Soundness:** 3
**Presentation:** 3
**Significance:** 2
**Originality:** 2
**Overall Recommendation:** 4
**Confidence:** 4

**Summary:**

The paper proposes RACO, a reward-free multi-objective alignment framework that resolves gradient conflicts among multiple DPO-style preference losses using a clipped variant of Conflict-Averse Gradient Descent (CAGrad-Clip). The authors prove convergence to Pareto-critical points while respecting user-specified objective weights and establish a one-step descent “acceleration” result for the two-objective case.

**Compliance With Llm Reviewing Policy:**

Affirmed.

**Final Justification:**

I tend to have a declining attitude towards acceptance.

**Key Questions For Authors:**

1.Could you provide full training and implementation details (optimizer, LR, batch size, gradient clipping norms, number of steps/epochs, LoRA vs. full updates, decoding parameters) and report multi-seed means/variances to assess statistical robustness?

2. What judge models were used for each task (particularly the reference to “GPT-5.1”)? Did you perform cross-judge evaluations or spot human assessments to validate Pareto dominance claims?

3. Can you share diagnostics on gradient conflicts (cosine similarities, magnitude ratios) across training, and how clipping changes these distributions?

4. For the quality–conciseness setup, conciseness is defined by length. Do your conclusions hold with alternative conciseness proxies (e.g., brevity penalties or compressibility) and under generation-based evaluations rather than margin-only proxies?

5. How does RACO scale with m>2 objectives in practice (runtime, stability)?

6. Could you report results on a 3–4 objective setting (e.g., quality/faithfulness/conciseness/safety) and discuss any changes to the clipping rule?

7. How does RACO compare empirically to conflict-aware gradient methods such as MGDA, PCGrad, and AUPGrad on your benchmarks? If not included, what prevented fair comparisons?

**Strengths And Weaknesses:**

Strengths:
1. Evaluations on multiple objective pairs (quality–conciseness; quality–faithfulness; helpfulness–harmlessness) and across multiple LLM families (Qwen, Llama, Gemma; both instruct and base variants) increase external validity.

2. The approach is easy to integrate into existing DPO-style pipelines and likely broadly useful for practitioners seeking controllable trade-offs.

3. The theoretical statements are easy to parse; the Pareto-criticality measure is standard and helps interpret guarantees.

Weaknesses:

1.Scalability to more than two objectives is not well explored empirically; theoretical results hold in general m, but practice is demonstrated mainly for m=2 with one m=3 mention in summarization (implicitly), and the overhead/behavior for larger m is unclear.

2. Heavy reliance on LLM-as-a-judge evaluations (including a reference to “GPT-5.1”) without cross-judge robustness checks, human evaluation, or inter-judge agreement raises concerns about metric reliability and reproducibility.

3. The “heuristics” beyond clipping (claimed to “improve our method”) are not concretely enumerated in the main text.

4. Multi-objective gradient surgery and projection-based methods (MGDA, PCGrad), and recent aggregators such as AUPGrad are not experimentally compared despite being highly relevant; the related work section could better reconcile how RACO differs from and improves upon these.

---

> ### Author Rebuttal · Authors · 2026-03-28
>
> We genuinely appreciate these detailed and constructive feedbacks. All further experimental results you requested are in **rebuttal package: <https://rebuttal-paper2.tiiny.site>**.
>
> ---
>
> **W1 & Q5 & Q6: Further experiments & implementation for $m>2$**
>
> We now extend both the solver and experiments to higher dimensions.
>
> We scale efficient solvers beyond the original $m=2$ case and use them to run experiments for 3-objective setting. This strengthens the scalability in practice, together with the general convergence guarantee already given in the paper.
>
> In rebuttal package, Section E.1 reports Pareto frontiers measured by both final judge scores and final validation margins, together with examples of in-training performance. Figures 7--8 show that RACO maintains a more favorable Pareto frontier under both metrics, and Figure 9 further shows that, similar to the $m=2$ case, RACO still improves less-preferred objectives and achieves a better overall trade-off.
>
> For runtime, thanks to the efficient solver, RACO remains efficient: training over 90K pairs only takes 15 min more from $m=2$ to $m=3$ due to additional objective loss calculation. We also clarify that there is no change to the clipping rule in the higher-dim setting, and we further provide a systematic clipping ablation in Section E.2.
>
> ---
>
> **Q1: Implementation details & multi-runs**
>
> We clarify that most training details (e.g., batch size & decoding config) are already provided in Appendix C.2 and C.3, and the full hyperparameters are included in supplementary README.md. We will add the remaining implementation details (e.g., AdamW and full-para finetuning) to the manuscript.
>
> Regarding statistical robustness, our setting is fully offline and doesn't involve online sampling or reward-model training, so the optimization is typically stable across run. Still, to address this concern directly, we provide multi-seed (5 per run) results with min-max bands for the corresponding final scores and in-training validation curves in Section E.4 of the rebuttal package.
>
> ---
>
> **W2 & Q2: Heavy reliance on LLM-as-a-judge**
>
> We apologize for the confusion, and clarify that **only Table 2 uses LLM-as-a-judge evaluation**, following the official BeaverTails win-rate setup. As stated in Appendix C.2 and C.3, the other evaluations use official released scoring models (trained encoder/decoders) from the benchmarks or standard community practice from prior works.
>
> We have revised the evaluation setup paragraph to make this distinction more explicit. In addition, we also report validation-batch objective margins and human-assessing case studies in Appendix C.1, which provide complementary evidence from different perspectives.
>
> ---
>
> **W3: Heuristics beyond clipping**
>
> We apologize for the lack of clarity. Clipping is the only heuristic used in our method, and we have revised the abstract to make this explicit by replacing the vague phrase "some heuristics" with "clipping heuristic".
>
> ---
>
> **Q3: Diagnostics on gradient conflicts**
>
> We have added it to Section E.5. In short, clipping steers training toward a more balanced policy, resulting in more stable gradient-conflict patterns with reduced variance throughout training.
>
> ---
>
> **Q4: Conciseness in generation-based evaluation**
>
> Our conclusions hold beyond margin-based proxies. In Fig 8 of Section E.1, we report generation-based conciseness score, measured by the length gap between the model-generated response and the gold preferred concise response provided in the dataset. The observed trend is consistent with the margin-based evaluation.
>
> ---
>
> **W4 & Q7: Comparison to broader multi-gradient methods**
>
> We agree that methods such as MGDA, PCGrad, and UPGrad should be further discussed. Our reason for building on CAGrad is: MGDA and PCGrad stop as soon as they reach the Pareto set and do not respect user-specified weights. By contrast, CAGrad converges to a stationary point of a user-specific weighed objective which is Pareto critical. This property is what we need in personalized alignment with static input weights. Moreover, the CAGrad paper showed that CAGrad outperform MGDA and PCGrad (e.g., CAGrad achieves the best average result on NYU-v2 and CityScapes, along with speedup over methods for multi-task RL).
>
> We have added all of the methods you mentioned to the related work section and expanded the discussion to make the distinctions and rationale more explicit. In summary, MGDA, PCGrad, and UPGrad are general-purpose and integrating them fairly into our setting requires additional adaptation to specific preference losses and weighted alignment. We also agree that adding broader comparisons would strengthen the paper and we will position them more clearly as valuable future work.
>
> ---
>
> We again appreciate your close reading and detailed feedback, and we look forward to an open-minded discussion if any such concerns remain.
>
> Sincerely,
>
> Authors of Paper 2

---

> > ### Author Rebuttal · Reviewer_yLW1 · 2026-04-04
> >
> > Thanks for your responses. I will change the final score to 4.

---

> > > ### Author Response · Authors · 2026-04-04
> > >
> > > Dear Reviewer yLW1,
> > >
> > > We genuinely appreciate your time for providing these constructive and detailed feedbacks to help us improve the clarity and quality of the manuscript.
> > >
> > > Again, thank you for supporting our work.
> > >
> > > ---
> > >
> > > Sincerely,
> > >
> > > Authors of Paper 2

---

### Official Review · Reviewer_3LnL · 2026-03-12

**Soundness:** 3
**Presentation:** 3
**Significance:** 3
**Originality:** 2
**Overall Recommendation:** 5
**Confidence:** 3

**Summary:**

This paper addresses the issue of conflicting alignment feedback to optimize LLMs. The paper explains that direct alignment methods are used to optimize models  with human preferences (RLHF, DPO, etc.), but that the data and the real world may contain conflicting feedback on what is preferred for most inputs. They propose a reward-free alignment method that directly uses pairwise preference data and resolves gradient conflicts via clipping. They also improve the method using heuristics. They present experiments using 3 LLM families (Qwen 3, Llama 3, Gemma 3), and they compare it with other reward free and preference weight input alignment methods like MODPO and AMOPO. They call their method RACO.

**Compliance With Llm Reviewing Policy:**

Affirmed.

**Final Justification:**

The reviewers added additional experiments that i think raised the value of the experiments and the paper.

**Key Questions For Authors:**

For models already finetuned, are the key capabilities of the models lost after applying your method? or in other words, would it make sense to add a regression test to see if the model is able to distinguish between the two conflicts but also whether the main original objectives are not lost? Your evaluation is limited to the two conditions being assessed in this dataset, but in alignment it is key to know if the model suffers from reward hacking or any other alignment or forgetting issues.

**Limitations:**

Yes they have. The authors discussed that they used public benchmarks, and that they did not collect any additional data beyond what is published elsewhere (they also used the LLM judges provided in those benchmarks).

**Strengths And Weaknesses:**

Strenghts:

1. The paper is well written and it is easy to follow. The idea is also clever and they compare with the relevant previous work in the same direction.

2. The paper includes multiple alignment frameworks: reddit and safety alignmnent, which shows that the approach generalizes beyond a single dataset.


Weaknesses:

1. all experiments are on aligning the model on one dataset (both for models already aligned Reddit, and for "base" models, as in the Beavertails experiments). I wonder if the authors could consider aligning one of the base models on an actual alignment dataset that may contain all desired phenomena, not just two conditions.

2. It would be good to study other types of conflicts, like helpfulness vs truthfulness, style vs truthfulness, refusal vs user  satisfaction, etc. This would help to show that the model is actually leading to the best desired setting in additional settings. 2 datasets are just limited to show the desired setting.

---

> ### Author Rebuttal · Authors · 2026-03-28
>
> We sincerely appreciate you for your time to provide these detailed and insightful points. We clarify each point below and provide further experiment results in the **rebuttal package: <https://rebuttal-paper2.tiiny.site>.**
>
> ---
>
> **W1: All experiments are on aligning the model on one dataset with only two conditions**
>
> We agree that evaluating on richer alignment settings is important. To address this, we now include additional $m=3$ (with 3 objectives) experiments on the Reddit task in the rebuttal package. Section E.1 reports Pareto frontiers measured by both final judge scores and final validation margins, together with examples of in-training performance. Figures 7 and 8 show that RACO maintains a more favorable Pareto frontier under both evaluation metrics, while Figure 9 further illustrates that, similar to the $m=2$ case, RACO can still improve less-preferred objectives and achieve a more balanced overall trade-off in the presence of conflicting objectives.
>
> Section E.2 also includes a systematic clipping ablation in the higher-dimensional setting, further supporting the necessity of clipping beyond the two-objective case. We provide detailed setup information in the figure captions. In addition, runtime remains practical: on roughly 90K pairs, the $m=3$ setting adds only about 15 minutes compared to $m=2$. This is because that we extend the efficient Line 5 solver in Algorithm 1 beyond the original $m=2$ case in Appendix B.1 and use it in the new experiments.
>
> We also hope you understand the specific scope of this work: our setting studies multi-objective alignment with **personalized weight input**, so moving to four or more objectives would require sweeping a much larger weight simplex in order to plot the Pareto frontiers, which quickly becomes computationally prohibitive. In this sense, our new $m=3$ results substantially broaden the original empirical scope. This is also broadly consistent with prior multi-objective alignment work we discussed, where most papers focus primarily on $m=2$, and only occasionally include $m=3$ studies (e.g., Appendix B of MODPO, ACL 2024).
>
> Furthermore, we note that the main target of this work is to resolve **conflicting** objectives. More realistic alignment datasets may involve many phenomena, but not all of them are strongly conflicting in the optimization sense. The additional experiments is intended as a meaningful intermediate step toward such richer settings, while still preserving the core question studied in our paper.
>
> ---
>
> **W2: It would be good to study other types of conflicts, 2 datasets are just limited to show the desired setting.**
>
> We understand this concern and agree that evaluating additional conflict types would further strengthen the paper. Our experimental design reflects a trade-off under limited computational budget: instead of testing many conflict pairs on a narrow set of models, we chose to evaluate across multiple recent model families (Qwen3, Llama3, Gemma3), including both base and instruction-tuned variants, to better demonstrate the broad applicability of RACO. This also required re-running all baselines on the same modern backbones and tuning their hyperparameters carefully for fair comparison, where we provide the full list of optimized hyperparameters and sweep ranges for all methods across different model families in the supplementary code's README.md file.
>
> We also note that, because this work introduces a new clipped-CAGrad method, we devoted substantial budget to systematic ablations, which are necessary to isolate the effect of clipping and support the main claims. For these reasons, we selected two representative and widely used benchmarks that capture strong objective conflict in summarization and safety alignment.
>
> That said, following your suggestion, we will work over more experiments in the further conflicting pairs you suggested during April and aim to add them to the final version.
>
> ---
>
> **Q1: Are the key capabilities of the models lost?**
>
> We interpret it as whether training on two objectives degrades performance on a third, untrained objective. We believe this is exactly where RACO is useful: as already suggested by Figure 2(a), existing methods tend to over-optimize the heavily weighted objective and sacrifice the less-preferred one, whereas RACO finds a more balanced update direction.
>
> Following the setup of Figure 2, we now evaluate two-objective trained models on a third held-out objective in Section E.6. We find that, while RACO already achieves the best frontier on the trained objectives, it also better preserves performance on the untrained objective.
>
> ---
>
> We again appreciate your close reading and insightful feedback, which we believe has significantly strengthened the evaluation and overall significance of RACO. We hope that our responses have addressed your concerns, and we look forward to an open-minded discussion if any such concerns remain.
>
> Sincerely,
>
> Authors of Paper 2

---

> > ### Author Rebuttal · Reviewer_3LnL · 2026-04-01
> >
> > The authors responded to my questions and added additional experiments justifying it.

---

> > > ### Author Response · Authors · 2026-04-01
> > >
> > > Dear reviewer 3LnL,
> > >
> > > Again, we sincerely appreciate for your time and effort in reviewing the manuscript and feedbacks to guide us to improve the manuscript.
> > >
> > > In the end, thank you for your support and valuable input towards our work.
> > >
> > > ---
> > >
> > > Sincerely,
> > >
> > > Authors of Paper 2

---

### Official Review · Reviewer_aXbp · 2026-03-12

**Soundness:** 3
**Presentation:** 3
**Significance:** 3
**Originality:** 3
**Overall Recommendation:** 4
**Confidence:** 3

**Summary:**

This paper targets a concrete and important issue in reward-free multi-objective alignment: preference gradients from different objectives can conflict, and naive weighting can produce unstable or distorted updates. The proposed fix is straightforward but meaningful: adapt CAGrad to this setting and add clipping so that the correction does not overemphasize lower-priority objectives.

**Compliance With Llm Reviewing Policy:**

Affirmed.

**Final Justification:**

The authors addressed my concerns, so I am maintaining my positive score.

**Key Questions For Authors:**

1. Add a more aggregated comparison across weight settings.
2. Discuss how the framework would handle more realistic multi-dimensional annotation formats.
3. Expand the discussion of scalability and stability in many-objective settings.

**Limitations:**

see Questions.

**Strengths And Weaknesses:**

Strengths:
1. The problem is clearly defined and important.
2. CAGrad-Clip is a simple but well-motivated modification rather than an unnecessarily complicated new framework.
3. The theory is meaningful: convergence to Pareto-critical points and the acceleration result in the two-objective case are both valuable.

Weaknesses:
1. Under more balanced weight settings, the gain over baselines is less pronounced. A more aggregated summary across weight configurations would make the empirical case stronger.
2. The method assumes objective-specific preference data. That is fine for the benchmarks, but real applications often collect multi-dimensional judgments from the same annotator, and the paper does not discuss this setting enough.
3. The hyperparameter `c` still needs model-family-specific tuning.
4. The strongest acceleration guarantee is only given for the two-objective case; for larger numbers of objectives, the support is mostly empirical.

---

> ### Author Rebuttal · Authors · 2026-03-28
>
> We genuinely appreciate your time to provide these insightful and helpful feedbacks. According to your suggestions, we have added further experiment figures and tables, all results are presented in the **rebuttal package: <https://rebuttal-paper2.tiiny.site>.**
>
> ---
>
> **W1 & Q1:  More aggregated weight summary**
>
> We have added an aggregated summary in Section E.3 of the rebuttal package. In Table 3, we report aggregated relative improvement over the pre-alignment model, averaged across model setups at each weight, together with the overall mean across the full weight sweep.
>
> We agree that under some more balanced weight settings, the gap can be less pronounced than in highly conflicted regimes, such as the Reddit task at weight (0.5, 0.5). This behavior is task-dependent and is naturally influenced by the degree of gradient conflict and the local optimization geometry. However, balanced weights do not necessarily diminish RACO's advantage: for example, on BeaverTail at weight (0.5, 0.5), RACO still achieves a substantially stronger average gain than both AMoPO and DPO LW (+1.312 versus +1.009 and +0.965, respectively).
>
> Overall, we think the new aggregated table better reflects the main strength of RACO: consistent improvement across the full range of user-specified weights, rather than only at a few isolated operating points. Thanks for raising this important point.
>
> ---
>
> **W2 & Q2: preference data structure**
>
> We apologize for any confusions and clarify that the dataset already has joint multi-objective annotations for each response pair: for every prompt-response tuple $(x, y^a, y^b)$, we have the annotation indicating whether $y^a$ wins $y^b$ or $y^b$ wins $y^a$ under every objective $i$. In other words, our method does not assume separate datasets for different objectives. We have revised Lines 166-172 to make this explicitly clear and avoid further ambiguity. Thank you for pointing this out.
>
> ---
>
> **W3: $c$ needs model-family-specific tuning**
>
> We agree that the correction radius $c$ is a hyperparameter that may require tuning across model families, and we have added this point to the new limitations section. Still, we would like to clarify that this issue is not unique to RACO, but is inherited from the underlying CAGrad mechanism itself, since $c$ controls the aggressiveness of conflict correction. The appropriate value naturally depends on the geometry and scale of the gradients, which can vary across model families and initialization regimes.
>
> Meanwhile, our ablations suggest that performance is reasonably stable within a local range (e.g., around $c=0.4$ for Qwen3 in Figure 6b), indicating that the method is not overly brittle in practice and $c$ can be controlled during finetuning. We believe this also opens an interesting direction for future work: developing a adaptive scheme for adjusting $c$ during training, e.g., based on gradient-conflict statistics, we will add a discussion section for this point.
>
> ---
>
> **W4 & Q3: Expand the discussion of scalability and stability in many-objective settings**
>
> We walk through the further results over more objective settings ($m>2$).
>
> On the theoretical side, we previously provided an efficient solution for line 5 of algorithm 1 only for $m=2$ ($\S$B.1), but we now derive the same efficient solution for higher dimensional objectives and use it to conduct new experiments. This ensures that the lightspeed implementation can scale beyond $m=2$ case.
>
> Regarding the acceleration guarantee for clipping, we agree that our strongest formal result currently applies only to the $m=2$ case. We admit that extending this guarantee to general $m$ would likely require new technical ideas in the scale of a new paper, and we view this as a promising direction for future theoretical work rather than a minor extension of the present analysis. A new technical discussion over general $m$ acceleration has been added as a remark.
>
> On the empirical side, we additionally include new 3-objective experiments in the rebuttal package. Section E.1 reports Pareto frontiers measured by both final judge scores and final validation margins, together with examples of in-training performance. In Figure 7 and 8, RACO maintains a more favorable Pareto frontier under both evaluation metrics. Figure 9 further illustrates that, similar to the $m=2$ case, RACO can still improve less-preferred objectives and achieve a more balanced overall trade-off in the presence of conflicting objectives. Section E.2 also includes a systematic clipping ablation in the higher-dimensional setting. We provide detailed setup information in the figure captions.
>
> ---
>
> We again appreciate your close reading and valuable feedback, which we believe has significantly strengthen the significance and advantage of RACO. We hope that our responses have addressed your concerns, and we look forward to an open-minded discussion if any such concerns remain.
>
> Sincerely,
>
> Authors of Paper 2

---

> > ### Author Rebuttal · Reviewer_aXbp · 2026-04-04
> >
> > Thank you for addressing my concerns. I will maintain my positive rating

---

> > > ### Author Response · Authors · 2026-04-04
> > >
> > > Dear Reviewer aXbp,
> > >
> > > Again, We sincerely appreciate your time in providing these insightful suggestions that help us to improve our work.
> > >
> > > In the end, thank you for your support to us.
> > >
> > > ---
> > >
> > > Sincerely,
> > >
> > > Authors of Paper 2

---

### Official Review · Reviewer_QkLL · 2026-03-21

**Soundness:** 3
**Presentation:** 4
**Significance:** 3
**Originality:** 3
**Overall Recommendation:** 5
**Confidence:** 3

**Summary:**

Most existing direct alignment methods are inherently single objective methods even though many alignment problems involve multiple conflicting objectives, for example harmlessness and usefulness. This paper proposes a framework for multi-objective LLM alignment without explicit reward modelling that resolves gradient conflicts by clipping the Conflict Averse (CA) gradient descent. They provide convergence guarantees to Pareto-critical points that respect user-specified objective weights and also show that clipped version strictly improves the convergence rate for the case with two objectives. Experiments also show improved performance when compared to existing multi objective reward free methods.

**Compliance With Llm Reviewing Policy:**

Affirmed.

**Key Questions For Authors:**

Q1) Check the question on SGD and sample complexity in limitations review. Adding a result on this would make this a stronger submission.

**Limitations:**

The paper would benefit from an explicit discussion of its limitations. It would be interesting to understand whether doing SGD would lead to a Pareto optimal point instead of it being just critical (which are locally optimal points rather than global). The paper provides optimization guarantees (convergence to empirical Pareto-critical points), but does not address whether these points generalize to the underlying population objectives. In particular, no sample complexity or statistical consistency guarantees are provided, leaving open the question of whether the learned trade-offs reflect true preference structures or artifacts of finite data.

**Strengths And Weaknesses:**

Strengths:

S1) Significance: In the context of multi-objective alignment (potentially being in conflict with each other which would likely be the case in a lot of real deployments) the CAGrad with clipping along with the theoretical guarantees is an important contribution. Understanding the behavior theoretically backed with the empirical experiments that show its effectiveness compared to the existing techniques make it impactful in the domain of alignment. Further that the framework does not involve explicit reward modeling makes it fast in practice and the work also shows that the framework integrates smoothly to LLM alignment.

S2) Presentation: Well written paper. Easy to follow and enough details to understand its position in the context of prior work.

S3) Originality: Provides a detailed analysis of CAGrad with clipping that also performs empirically better than existing methods. This is a novel application of CAGrad to alignment and when the technique directly does not lead to the same guarantees clipping is introduced for the guarantees to go through. Moreover, this technique smoothly integrates for LLM alignment. This opens up directions for further work on understanding and guiding the behaviour of multi objective alignment.

S4) Soundness: Proofs look sound although I skimmed the proofs at a high level and did not check the derivations in detail. Might be better to clarify in the introduction that the strict improvement when clipping is active is for a single step. Experiments are designed to support the claims made.

Weakness:

W1) Originality - The work builds heavily on techniques and analyses from the CAGrad literature. While the adaptation is interesting and the clipping is new, the core methodology itself introduces limited technical novelty.

W2) Presentation - please add a ‘limitations’ subsection. See limitations review for further details.

---

> ### Author Rebuttal · Authors · 2026-03-28
>
> We sincerely thank you for your time to provide these detailed and insightful points. We clarify each point below:
>
> ---
>
> **1. The work is heavily based on techniques and analyzes from the CAGrad literature.**
>
> We agree that CAGrad is the main foundation of our method, and we will revise the paper to make that relationship more explicit. Our contribution is to adapt CAGrad to reward-free weighted multi-objective LLM alignment and to analyze the clipped variant that this setting requires. In particular, RACO works with objective-specific DPO losses, uses a user-weighted anchor gradient rather than the uniform average used in the original CAGrad paper, and then applies a new coefficient-clipping step before forming the correction direction. This clipping step is absent from the original CAGrad algorithm and is introduced to ensure that the conflict-correction term does not allocate more mass to a low-priority objective than the user-specified weights allow.
>
> We also agree that the theoretical distinction from CAGrad should be stated more concretely. The original CAGrad theorem is for the unclipped update and shows that the method converges to a stationary point of the average loss, which is also Pareto-stationary. Our analysis is still based on the descent argument but the one-step progress will change once clipping is introduced. In other words, our theorem is not a straightforward consequence of the existing CAGrad result. It is a new one for a different update rule and a different objective. Moreover, in the two-objective case, we prove that clipping yields a strictly stronger one-step progress than vanilla CAGrad. Indeed, clipping can move the correction direction closer to the user-weighted anchor gradient. This acceleration result does not appear in the original CAGrad paper. We will revise the paper to highlight these contributions more clearly.
>
>
> ---
>
> **W2 & Q1: Limitation Section & Question on SGD and sample complexity**
>
> We agree and will add an explicit limitations subsection. First, our current theory is a first-order optimization result for a nonconvex weighted empirical objective. Without additional structure, the best possible result that we hope for is the convergence to a Pareto-critical point, which has been shown in Theorem 3.1. This does not imply global Pareto optimality of the alignment problem in general. Second, our current proofs do not provide a stochastic-gradient theorem for minibatch training. Although Algorithm 1 is implemented with minibatches, the proof of Theorem 3.1 proceeds by applying a deterministic descent argument. A full SGD analysis would need to control not only the gradient variance, but also the sensitivity of the clipping map to that variance. We do not provide such a result and we will state this explicitly. Third, our paper focuses on designing efficient algorithms for offline alignment problems in which the weighted loss is induced by the **fixed preference dataset**. Understanding (i) how many objective-specific preference pairs are needed to recover a desired population trade-off and (ii) whether the empirical Pareto frontier converges to the population Pareto frontier is beyond the scope of this paper. Therefore, our current analysis does not provide sample complexity bounds, statistical consistency guarantees, or generalization guarantees. We will add these points explicitly to the limitations section so that the scope of our results is clear.
>
>
> ---
>
> **Additional experiment results**
>
> Suggested by the other reviewers, we have further conducted experiments over higher-dimension setting, with all results are presented in the **rebuttal package: <https://rebuttal-paper2.tiiny.site>.**
>
> In summary, we extend the original efficient $m=2$ solver to higher-dimensional objective spaces and provide new three-objective experiments in Section E.1 to further support the scalability of RACO, together with a systematic $m=3$ clipping ablation in Section E.2 demonstrating that clipping remains necessary in higher-dimensional multi-objective alignment.
>
> ---
>
> We again appreciate your close reading and valuable feedback, which we believe has significantly improved the clarity and scope of the paper focus. We hope that our responses have addressed your concerns, and we look forward to an open-minded discussion if any such concerns remain.
>
>
> Sincerely,
>
> Authors of Paper 2

---

> > ### Author Rebuttal · Reviewer_QkLL · 2026-04-03
> >
> > Thank you for addressing my concerns. I will maintain my positive rating

---

> > > ### Author Response · Authors · 2026-04-04
> > >
> > > Dear Reviewer QkLL,
> > >
> > > Thank you again for your time in providing these valuable input. We sincerely appreciate your support to our work.
> > >
> > > ---
> > >
> > > Sincerely,
> > >
> > > Authors of Paper 2

---

### Decision · Program_Chairs · 2026-04-30

**Decision:**

Accept (spotlight)

**Comment:**

The paper tackles the important problem of multi-objective alignment in LLMs, proposing a reward-free framework that resolves gradient conflicts via a clipped variant of conflict-averse optimization. In the initial reviews, reviewers appreciated the significance, clarity, and practical relevance of the method, while raising concerns regarding limited novelty, evaluation breadth, and missing discussion. During the rebuttal, the authors addressed the concerns well, which led all reviewers to mark their concerns as resolved and maintain or increase their scores (all positive ones). Overall, the work present a technically solid and practically useful contribution to multi-objective alignment, and given the strengthened empirical support and clarified scope after rebuttal, I recommend strong acceptance.